# Conscious Indirect Blood Pressure Measurements in Asiatic Black Bears (*Ursus thibetanus*)

**DOI:** 10.3390/ani16010146

**Published:** 2026-01-05

**Authors:** Grace M. Scrafford, O. Lynne Nelson, Rachel Sanki, Sarah van Herpt, David Rice

**Affiliations:** 1College of Veterinary Medicine, Washington State University, Pullman, WA 99163, USA; 2Vietnam Bear Rescue Center, Hồ Sơn, Tam Đảo National Park, Tam Dao 283600, Vĩnh Phúc, Vietnam; rachel.sanki@outlook.com (R.S.); svanherpt@animalsasia.org (S.v.H.); 3Center for Interdisciplinary Statistical Education and Research, Washington State University, Pullman, WA 99163, USA; david.rice@wsu.edu

**Keywords:** aortic aneurysm, Asiatic black bear, bear bile, blood pressure measurement, systemic hypertension, animal welfare, *Ursus thibetanus*

## Abstract

At Animals Asia’s Vietnam Bear Rescue Center, 40% of bears are diagnosed with systemic hypertension (high blood pressure in the arterial system) and require daily medication to manage it. Veterinarians have only been able to diagnose systemic hypertension in this population during health checks while animals are under anesthesia. Anesthetic drugs used have the potential to confound results of blood pressure measurements; for that reason, hypertension is diagnosed through validated secondary structural lesions. Recently, bears at the rescue center in Vietnam have been trained for cooperative conscious blood pressure measurements. Using direct arterial catheterization, this study validated indirect blood pressure measurements and then used this indirect method to determine normal ranges of blood pressure measurements in conscious, trained bears. This will allow for early diagnosis of systemic hypertension and improve the ability to evaluate protocols used for treatment. This knowledge will improve the quality of care for the hundreds of bears rescued from the bile industry and for other captive bears worldwide.

## 1. Introduction

Bear bile has been used in traditional Asian medicine for centuries. Overhunting, for the collection of bile, has contributed to significant declines in populations of Asiatic black bears (*Ursus thibetanus*) and sun bears (*Helarctos malayanus*), which are classified as vulnerable by the IUCN Red List of Threatened Species [1]. In more recent history, large-scale production operations have been developed in which live bears are subjected to extraction methods through catheters or fistulas directly into the gallbladder [2,3]. These extraction methods result in chronic pathologies including cholecystitis, hepatic neoplasia, and peritonitis [2,4,5]. Bears rescued from the bile industry have been found in studies to have a mean lifespan of 17 years, half that of expected of healthy bears in captivity [4]. Bears in these large-scale bile operations have been known to have severely reduced mobility and often lack nutrition and access to water [5,6]. The high stress, systemic infection, and increased incidence of renal disease are thought to be correlated with increased rates of systemic hypertension in bears rescued from bile extraction operations [2].

In 2005, Vietnam made bear bile possession, sale, and extraction illegal, and over the intervening years, an influx of animals has been relocated to rescue centers. At the Vietnam Bear Rescue Center (VBRC), 40% of Asiatic black bears rescued are afflicted with lesions associated with systemic hypertension, as assessed by veterinarian-documented, validated secondary structural changes (personal communication, VBRC veterinarians). Anesthetic drugs administered at health checks, including medetomidine and isoflurane, have been reported in other species to result in transient hypertension or hypotension. For that reason, diagnosis of systemic hypertension in this population is currently based on secondary structural criteria observed during examination, including retinopathies, left ventricular hypertrophy, and aortic dilation [2,7]. Since assessment of secondary lesions only occurs during anesthetized health checks, both diagnosing and monitoring the progression of systemic hypertension are difficult and inefficient, as many bears receive an anesthetized health examination every 1–2 years. Noninvasive and conscious blood pressure measurement techniques would be ideal to be able to identify and monitor systemic hypertension more frequently in captive bears.

Bears with lesions of systemic hypertension have shorter survival rates [2]. These bears require lifelong antihypertensive medications for this condition. In severe cases, chronic undiagnosed or uncontrolled systemic hypertension leads to aortic aneurysm and rupture at a relatively high rate in this population of bears. Cardiovascular disease in this population is the third leading cause of mortality [2]. It is vital that systemic hypertension is identified earlier, monitored, and treated for the well-being and prolonged survival of rescued bears in the sanctuary. The objective of this study was to evaluate a noninvasive method of blood pressure measurement in trained, cooperative Asiatic black bears and to determine normal active measurement (versus resting measurement) ranges for trained bears without lesions of systemic hypertension. A validated technique could then be used to monitor bears that are treated for lesions of systemic hypertension.

## 2. Materials and Methods

### 2.1. Blood Pressure Validation

The study was conducted using ARRIVE protocols for animal studies. The subjects of this study were Asiatic black bears (*Ursus thibetanus*) housed at Animals Asia’s VBRC. Nine bears, 6 males and 3 females with age ranges of 13–22 years, undergoing routine health examinations, were used to validate the oscillometric blood pressure measurement. Under general anesthesia, lingual arterial catheterization (heparinized; angiocatheter) was performed for direct blood pressure measurement, while simultaneously, indirect forelimb measurement was taken using oscillometric methods with a Mindray multiparameter Imec8 machine (Mindray Bio-Medical Electronics Co., Ltd. Nanshan, Shenzhen 518057, China), the same as used during conscious measurements [8]. The cuff width used in each individual was determined based on 30–40% of the forearm circumference (between the carpus and elbow) at the site of cuff placement. Measurements of systolic, diastolic, and mean arterial pressure were taken with each method (direct and indirect) at three- or five-minute intervals for a total of six measurements.

### 2.2. Statistical Analysis

Pearson correlation coefficient of systolic blood pressure measurements was calculated using direct and indirect values pooled from nine bears (Figure 1). On analysis, individual variability in correlation between systolic direct and indirect measurements was noted, and the Pearson correlation coefficient was also calculated using the mean of each individual bear’s systolic measurements (Figure 2).

### 2.3. Study Population

This study was a retrospective assessment of indirect blood pressure measurement with a clinical validation experiment in nine animals. Twenty-four rescued Asiatic black bears (*Ursus thibetanus*) were used to assess conscious blood pressure measurements. In the spring of 2021, at Animals Asia’s VBRC, staff began positive reinforcement training for cooperative conscious blood pressure measurements using an oscillometric indirect technique (Mindray multiparameter Imec8, Mindray, Mahwah, NJ, USA). Twenty-four bears were successfully trained, thirteen with structural lesions of systemic hypertension, diagnosed during annual anesthetized health checks by veterinary staff, and eleven without lesions of hypertension. Of the study population with lesions of systemic hypertension, seven were female bears, and six were male bears with an age range of 18–24 and an average age of 21 years. Of the study population without lesions of systemic hypertension, five were female bears and six were male bears with an age range of 7–23 and an average age of 15 years. Bears were trained to present a forelimb for placement of a blood pressure cuff on the antebrachium between the carpus and the elbow. The bears were trained using positive reinforcement with food rewards to position their forelimb at heart level, extended through den bars and into a cylindrical tube acting as a sleeve. In order to keep the forelimb extended and elevated, bears were trained to grab onto a bar at the end of the sleeve for cuff placement and acquisition of oscillometric measurements (Figure 3).

### 2.4. Experimental Design

Team members taking these conscious measurements were trained by and under the supervision of behavioral husbandry managers to follow a standard protocol designed with veterinary staff in order to promote consistency. The cuff width used was determined based on 30–40% of the forearm circumference at the site of cuff placement and was recorded in the individual animal’s medical record to be consistently used for subsequent measurements. The demeanor of bears, as well as any errors in measurements due to movement or machine error, were also recorded. Medical records from all twenty-four bears were reviewed along with all previously recorded indirect blood pressure measurements using this technique.

Once validated, indirect conscious blood pressure measurements were evaluated. Outliers within the sampling time, or errors in measurements, were determined using the interpretation of the bear’s demeanor, inconsistencies related to animal movement, or equipment error, which were recorded at the time of the measurement session. Systolic blood pressure measurements with a 50% or greater difference from the average were deemed inaccurate and were excluded. Ranges for conscious cooperative blood pressure measurements of bears with and without lesions of systemic hypertension were established, and the average was determined. Distribution of conscious cooperative blood pressure measurements of bears without lesions of systemic hypertension has been demonstrated (Figure 4).

### 2.5. Patient Example

An example of a 20-year-old female Asiatic black bear has been provided to demonstrate the use of conscious blood pressure measurements in an individual with suspected systemic hypertension (Figure 5). Three treatment protocols were prescribed at different time points. Protocol 1 (enalapril 0.5 mg/kg BID, amlodipine 0.2 mg/kg BID, and atenolol 0.5 mg/kg BID) was prescribed upon initial diagnosis and identification of severe structural lesions of systemic hypertension. Blood pressure measurements were taken 20 times over 204 days using the indirect method described above. Protocol 2 (enalapril 0.7 mg/kg BID, amlodipine 0.1 mg/kg BID, and atenolol 1 mg/kg) was initiated 7 months later after a recheck examination noted progressive lesions of systemic hypertension. Blood pressure measurements were taken 17 times over 264 days. Protocol 3 (enalapril 0.7 mg/kg BID, atenolol 1 mg/kg, with discontinuation of amlodipine) was initiated after adverse effects were noted with one of the medications (gingival hyperplasia). Blood pressure measurements were taken 14 times over 239 days.

## 3. Results

The indirect blood pressure technique using the Mindray multiparameter Imec8 was well correlated with direct arterial blood pressure measurements in Asiatic black bears. A Pearson correlation coefficient of 0.90 (95% CI: 0.84–0.94, *p*-value < 2.2 × 10^−16^) was found between pooled indirect and direct measurements from nine bears (Figure 1). By taking the mean of six systolic measurements in each bear, individual variability in correlation between indirect and direct measurements was corrected (Figure 2). The Pearson correlation coefficient of the mean of individual measurements was 0.93 (95% CI: 0.70–0.98, *p*-value = 0.00025), which was significant.

Asiatic black bears without lesions of systemic hypertension had indirect systolic measurements with a mean of 180.65 +/− 37 mmHg (95% CI: 126–255) (Figure 4). Bears with lesions of systemic hypertension receiving antihypertensive therapy had similar indirect systolic measurements of 189.49 +/− 40 mmHg (quantile 2.5%, 0.975%: 125–255). These values were established using conscious cooperative blood pressure measurements with the indirect technique (Mindray multiparameter Imec8). Blood pressure measurements reviewed from the medical records of hypertensive bears were all taken after the initiation of antihypertensive treatment. As a result, values for bears with lesions of systemic hypertension prior to medical therapy were not available.

An example of a single Asiatic black bear has been provided to demonstrate the use of conscious blood pressure measurements to assess medical therapy in an individual with suspected systemic hypertension (Figure 5). Protocol 1 displayed a relatively constant trendline with regard to systolic blood pressure measurements. In Protocol 2, dosages of two of the antihypertensive medications (enalapril and atenolol) were increased. This change in medication resulted in a decreasing trendline of blood pressure measurements. The antihypertensive drug, amlodipine, was discontinued in Protocol 3 due to adverse effects (gingival hyperplasia). Eliminating amlodipine resulted in an increasing trendline of systolic blood pressure.

## 4. Discussion

Indirect Doppler blood pressure measurement techniques have been validated in other bear species for use in conscious, trained animals [7]; however, the present study validated the indirect oscillometric technique against direct measurements with lingual arterial catheterization at the VBRC in nine adult Asiatic black bears undergoing routine health checks. The intent was to clinically use this indirect method in the facility. Using pooled systolic blood pressure measurements from all nine bears, we observed a strong positive correlation between indirect oscillometric and direct systolic measurement techniques. The strong correlation between the indirect and direct systolic blood pressure measurements supports the continued use of indirect oscillometric measurements for monitoring blood pressure in Asiatic black bears at the VBRC. While individual bears had more variation in correlation between direct and indirect techniques, the mean systolic blood pressure measurements in individual bears showed a robust correlation coefficient of 0.93 (CI 95%: 0.70–0.98). This information suggests that for the most accurate results, using indirect techniques, clinicians should average a series of multiple measurements from a patient. This finding aligns with the recommendations of the ACVIM consensus statement on the identification, evaluation, and management of systemic hypertension in dogs and cats. This consensus recommends discarding the first measurement and performing an average of 5–7 consistent indirect measurements when evaluating for systemic hypertension [9]. Isolated diastolic hypertension is not described in animals; thus, systolic blood pressure is typically the measure used in veterinary medicine as an indicator of systemic hypertension and was therefore the preferred measurement evaluated in this study.

Conscious cooperative blood pressure measurements reviewed from the medical records of hypertensive bears were all taken after the initiation of antihypertensive treatment due to significant structural lesions. Unfortunately, average systolic blood pressure ranges, or cut-offs for bears suspected to have systemic hypertension, could not be determined in this study. As this technique becomes routinely used, a cut-off or range of systolic blood pressure associated with lesions of systemic hypertension may be considered as an objective for future research. This would benefit veterinarians in diagnosing and determining when to initiate treatment for systemic hypertension in this population, prior to the identification of secondary organ damage.

It is important to note that in captive wildlife, training and performance of these measurements are often stimulating and exciting, particularly for bears, as they are highly motivated by food rewards. The indirect blood pressure ranges reported here should be considered ranges observed at an activity level in Asiatic black bears as opposed to a resting level. Animals with a playful demeanor differ in cardiovascular parameters from calm animals, which is not associated with stress [10]. Although training for cooperative indirect measurements requires staff time, it is important to note the benefits of cooperative husbandry on animal welfare. Previous studies have demonstrated a marked decrease in anxiety in bears trained for cooperative care compared to bears undergoing chemical immobilization for routine husbandry procedures [11].

A goal for the use of this data is the monitoring of clinical response to antihypertensive medication protocols. Monitoring conscious cooperative measurements in bears receiving antihypertensive medications would allow veterinarians to more frequently modify dosages and protocols in order to achieve adequate management of systemic hypertension. An example of monitoring the effect of changing treatment protocols was created in one bear with an average of seventeen indirect oscillometric systolic measurements in each of three drug protocols (Figure 5). Protocol 1 displayed a relatively constant trendline. In Protocol 2, dosages of antihypertensive medications, enalapril and atenolol, were increased due to the progression of lesions identified at an anesthetized health check. This change in dosage of medication resulted in a decreasing trendline. The antihypertensive drug, amlodipine, was discontinued in Protocol 3 due to adverse effects. In Protocol 3, there is an increasing trendline after eliminating one of the antihypertensive medications. Noting this response, the veterinary team is alerted to make additional medication adjustments. Antihypertensive treatment should be based on the trends of individual patients’ response, and it is hoped that this information will allow veterinarians to modify individual treatment protocols earlier in treatment to minimize the secondary organ damage resulting from systemic hypertension.

## 5. Conclusions

Indirect oscillometric blood pressure measurements (Mindray multiparameter Imec8) are a valid technique for assessing blood pressure in Asiatic black bears at the VBRC. The strong correlation between the indirect and direct, for systolic measurements, supports the continued use of indirect measurements for monitoring blood pressure in Asiatic black bears. Similar validation and monitoring could be important tools for captive wildlife that are at risk for systemic hypertension.

Trained and active Asiatic black bears at the VBRC without lesions of systemic hypertension had a mean indirect systolic blood pressure of 180.65 +/− 37 mmHg (95% CI: 126–255). It is important to note that this value in working bears is not considered a resting blood pressure measure.

## Figures and Tables

**Figure 1 animals-16-00146-f001:**
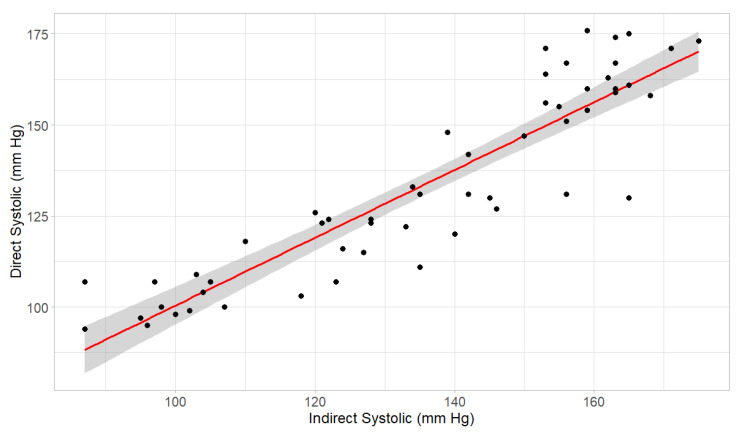
Correlation of systolic blood pressure measurements using direct and indirect techniques pooled from nine Asiatic black bears (*Ursus thibetanus*). Points represent the correlation between both measurement techniques used simultaneously in a patient. The Pearson correlation coefficient of 0.90 indicated a strong positive relationship.

**Figure 2 animals-16-00146-f002:**
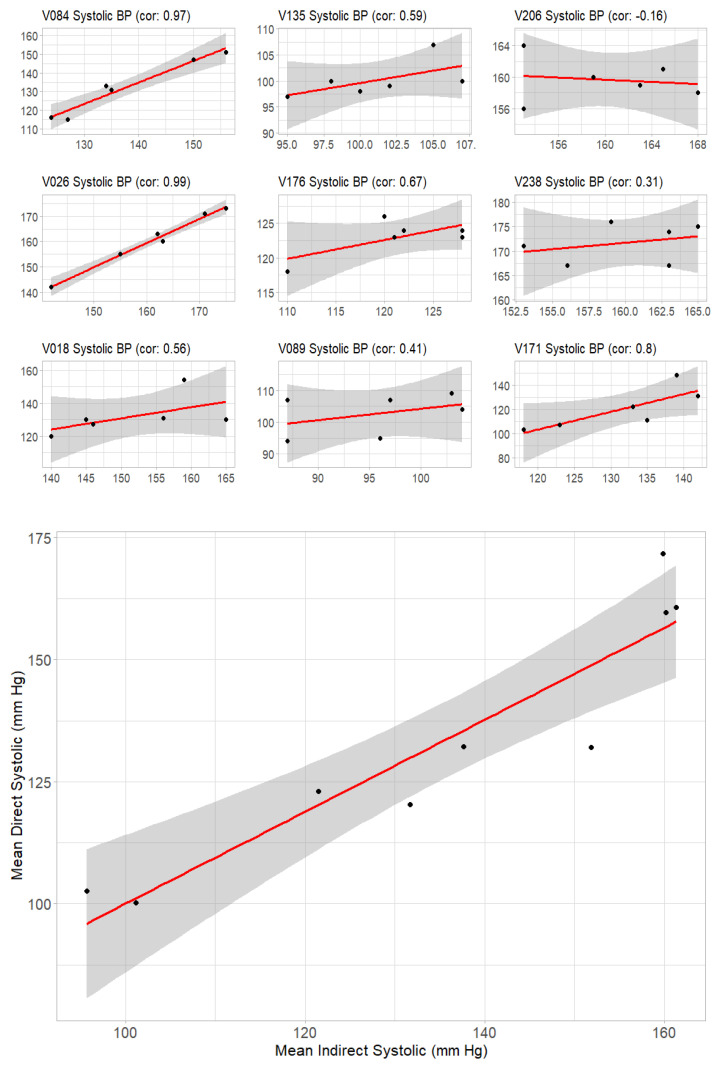
Correlation of systolic blood pressure measurements using direct and indirect techniques in nine Asiatic black bears (*Ursus thibetanus*). Correlation graphs made for simultaneous indirect and direct measurements in each bear reveal individual animal variability, while comparing the mean of the individual measurements shows a strong positive relationship. Pearson correlation coefficient of 0.93 (95% CI: 0.70–0.98).

**Figure 3 animals-16-00146-f003:**
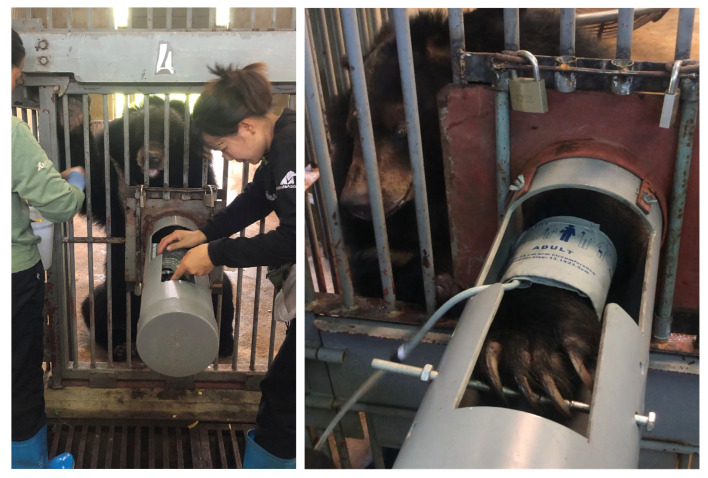
Asiatic black bear (*Ursus thibetanus*) at the Vietnam Bear Rescue Center, providing a forelimb for cooperative conscious blood pressure measurements using an indirect oscillometric technique.

**Figure 4 animals-16-00146-f004:**
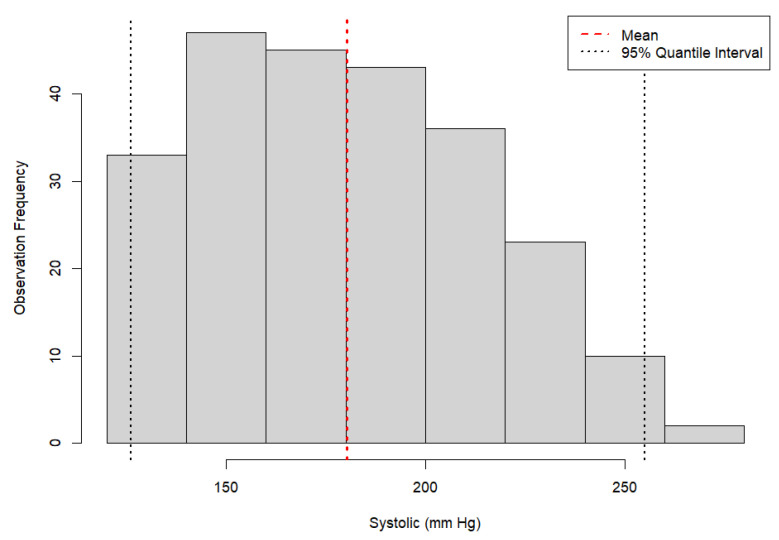
Distribution of indirect systolic blood pressure measurements taken in eleven conscious Asiatic black bears (*Ursus thibetanus*) without lesions of systemic hypertension. Measurements demonstrate a normal distribution around the mean of 180.65 mmHg.

**Figure 5 animals-16-00146-f005:**
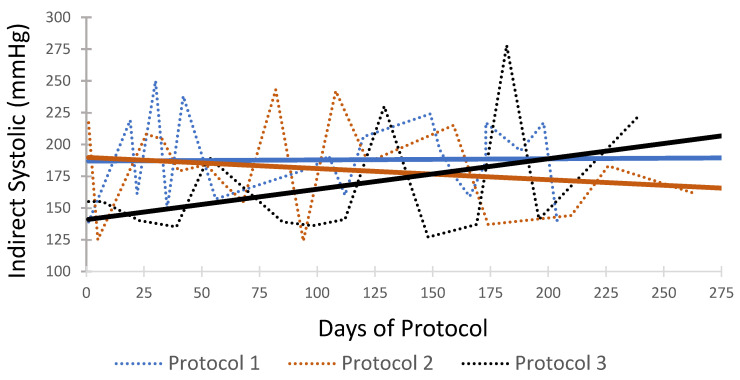
An example of the use of indirect measurements to monitor antihypertension treatment protocols in a rescued Asiatic black bear (*Ursus thibetanus*). Indirect blood pressure measurements were recorded through three different antihypertensive medication protocols. Protocol 1: Enalapril 0.5 mg/kg BID, Amlodipine 0.2 mg/kg BID, and Atenolol 0.5 mg/kg BID. Protocol 2: Enalapril 0.7 mg/kg BID, Amlodipine 0.1 mg/kg BID, and Atenolol 1 mg/kg. Protocol 3: Enalapril 0.7 mg/kg BID and Atenolol 1 mg/kg BID. Solid lines represent systolic blood pressure trend lines for each protocol.

## Data Availability

Data is unavailable due to privacy reasons.

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
