# Peer review of "Conscious Indirect Blood Pressure Measurements in Asiatic Black Bears (Ursus thibetanus)"

_animals, 2026, doi:10.3390/ani16010146_

Round 1

Reviewer 1 Report

Comments and Suggestions for Authors

Title: Conscious Blood Pressure Measurements in Asiatic Black Bears (Ursus thibetanus) with and without Lesions of Systemic Hypertension

  1. General Comments

This report aimed to validate an indirect method of blood pressure measurement in trained, cooperative Asiatic black bears and to determine normal active-measure (versus resting-measure) ranges for trained bears without lesions of systemic hypertension. The ultimate goal was to develop a validated technique to monitor bears treated for lesions associated with systemic hypertension.

The study is original for this species and provides a new perspective on the treatment of hypertension in an endangered bear population rescued from deplorable, inhumane conditions.  However, a few critical issues must be addressed for publication.

  1. Title

One of the main issues in this report is that "values for bears with lesions of systemic hypertension prior to medical therapy were not available" (lines 130-131). Since no results were presented for bears with systemic hypertension-related lesions, the title is not accurate. Thus, my suggested title is: Conscious Blood Pressure Measurements in Asiatic Black Bears (Ursus thibetanus).

  1. Simple Summary

The Simple Summary effectively explains the study's purpose and findings, but the term "systemic hypertension". The Summary should be revised to fix a few punctuation errors, avoid abbreviations entirely, and explain the clinical implications in everyday terms.

At Animals Asia’s Vietnam Bear Rescue Center, 40% of bears are diagnosed with systemic hypertension (high blood pressure throughout the body) and require daily medication to manage it. Veterinarians have only been able to diagnose systemic hypertension in this population during health checks while animals are under anesthesia. Anesthetic drugs used have the potential to confound results of blood pressure measurements; for that reason, hypertension is diagnosed through validated secondary structural lesions. Recently, bears at the rescue center in Vietnam have been trained for cooperative conscious blood pressure measurements. Using the data collected, this study validated conscious indirect blood pressure measurements. This will allow for early diagnosis of systemic hypertension and further our ability to evaluate treatment protocols. This knowledge will improve the quality of care for the hundreds of bears rescued from the bile industry and for other captive bears worldwide.

  1. Abstract

The Abstract includes essential elements: background, objectives, methods, results, and conclusions, and is appropriate.

  1. Introduction

The Introduction is clear and appropriate.

  1. Materials and Methods

There are a few issues in this section.

The first and foremost is related to the study population:

  1. a) The authors report that "Twenty-four bears were successfully trained, thirteen with structural lesions of systemic hypertension, diagnosed at annual anesthetized health check by veterinary staff, and eleven without lesions of hypertension" (lines 88-90). However, no results are available for bears with structural lesions caused by systemic hypertension, as the authors explain (lines 128-131). Thus, these animals should not be included in the study population; the limitations of the measurements obtained from them must be included only in the Discussion.
  2. b) It is also confusing that the study population (without lesions of hypertension) is initially mentioned as eleven (line 90) and further described as nine individuals (lines 110-111, 117). It seems that nine bears without hypertension-related lesions were used for a comparison study (direct and indirect systolic blood pressure measurements, Figures 1 and 2), and eleven animals without hypertension-related lesions were employed for indirect systolic blood pressure measurements in conscious bears (Figure 3). This must be clarified in the text. Also, the characterization of the nine bears (age, sex) should be explicit.

The blood pressure conditioning technique described is a very important aspect of this study. I suggest that the authors include an image of the measurement being taken on a bear to clarify the method further.

The following sentence must be revised: "Indirect Doppler blood pressure measurement techniques have been validated in other bear species [6], however, this study validated the indirect oscillometric technique with direct measurements with lingual arterial catheterization at the VBRC in nine adult Asiatic black bears undergoing routine health checks" (lines 108-111). My suggestion is: "Indirect Doppler blood pressure measurement techniques have been validated in other bear species [6]; however, the present study validated the indirect oscillometric technique with direct measurements with lingual arterial catheterization at the VBRC in nine adult Asiatic black bears undergoing routine health checks". Also, this type of sentence fits in the Discussion, not in the Materials and Methods Section, and should be moved.

  1. Results

The presentation of the results needs improvement.

In Figures 1 and 2, the correlation is more evident when a line of best fit is drawn. Please redo the graphs with the lines.

In both Figures 1 and 2 legends, the scientific name of the bears should be mentioned: "Correlation of systolic blood pressure measurements using direct and indirect techniques in nine Asiatic black bears (Ursus thibetanus)."

The authors report that "Asiatic black bears without lesions of systemic hypertension had indirect systolic measurements with a mean of 180.65 +/- 37 mmHg (95% CI: 126-255) (Figure 3). These values were established using conscious cooperative blood pressure measurements from thirteen bears with the indirect technique" (lines 140-143). As mentioned in the Materials and Methods, eleven bears (not thirteen) displayed no lesions of systemic hypertension (line 90). Please explain or correct this discrepancy.

Figure 3 must be revised and further explained. Since the Materials and Methods do not provide sufficient information on how the data were obtained (e.g., the total number of measurements), and the figure legend is not self-explanatory, the figure does not provide clear information. For instance, why does it show eight bars? What do the dotted lines indicate? Please elaborate.

As for Figure 4, it is not even mentioned in the brief description in the Results section. Furthermore, the procedure is not explained in the Materials and Methods section. How many animals were used for this experiment, for each treatment protocol? This should be described in the Materials and Methods and mentioned in the Results text.

Also, the following text is typical for the Discussion, since it is explaining the results: "The training and performance of these measurements is often stimulating and exciting for these bears and should be considered ranges at an activity level in Asiatic black bears as opposed to a resting level" (lines 143-146). It should be moved from the Results to the Discussion.

  1. Discussion

The Discussion is very brief and mentions only one reference. More references are necessary to increase the quality of the Discussion.

  1. Conclusion

The Conclusion is consistent with the results, particularly regarding the correlation between the indirect and direct systolic measurements.

  1. References

The authors listed three references (numbers 9, 10, and 11) that were not used in the manuscript. Please refer to them in the text or remove them.

Reviewer 2 Report

Comments and Suggestions for Authors

General comments

The early identification of conditions in animals destined for a particular area has become a challenge in veterinary medicine today, as it offers the opportunity for early care and the avoidance of systemic complications. Therefore, I consider this manuscript proposal to be appropriate and innovative. However, a significant weakness of your proposal is the lack of clarification regarding the method used to measure blood pressure, as this is important for assessing the measurement method's sensitivity.
Response:
Another significant weakness in your proposal is the lack of a clear and logical order in your methodology, as the experimental design and statistical analysis are mixed, making it difficult to understand how your work was carried out.
Response:
Finally, another weakness is the lack of a detailed description of the blood pressure measurement, especially specifying the equipment used for measurement.
Response:

Particular comments

Line 2. I agree with your proposed title; however, if the authors allow me, I suggest clarifying that it was a non-invasive measurement.
Response:
Lines 16-21. I understand that this section is intended to be a simple summary; however, you do not mention what your most relevant result that would allow it to be associated with the conclusion of your study.
Response:

Lines 24-35. These introductory sentences for your abstract are excessive. If the authors allow me, I suggest that they limit themselves to describing the importance of recognizing systemic hypertension in this species as a consequence of its zootechnical purpose. However, I also suggest that the objective of your study should be clarified in advance to ensure that it is consistent with your title, as I recommend caution with your phrase “Indirect blood pressure measures were validated with direct arterial measures,” since the validation of a method requires a different experimental design that necessarily uses the gold standard measurement method.
Response:

Line 36. If the authors allow me, they could indicate the demographic characteristics of the animals used. That is, the average age, sex, and average weight of the animals included in their study.
Response:
Line 43. I agree with your proposed keywords, but if you allow me, I suggest that you include the term “animal welfare.” This may increase the possibility of matching between different databases.
Response:
Line 45. In general, I find that this section is well structured, but references 2 and 4 are overused. I suggest replacing them with another reference that complements the section.
Response:

Line 53. This idea is relevant to the final consequence of the treatment administered to these animals. However, you could clarify that these pathologies are caused by chronic trauma generated during the extraction of bile fluid.
Response:
Line 63. Please add a reference.
Response:
Line 76. I agree with your description; however, I suggest that you indicate the gap in the information, for example, what is the importance of measuring blood pressure?
Response:

Lines 78-82. Based on my general comment, I suggest that you may want to rephrase the objective. For example, if the authors allow me, I suggest changing it to “the objective of this study was to evaluate noninvasive blood pressure measurement in Asiatic black bears with or without lesions of systemic hypertension.”
Response:
Line 83. In general, this section is not very clear. Based on my general comment, I suggest that this section be divided into subsections, for example, animals, experimental design, blood pressure measurement, statistical analysis, and, above all, the ethical statement indicating that the study was conducted using ARRIVE protocols for animal studies.
Response:

Line 84. I suggest that at this level you indicate the characteristics of your study, i.e., whether it was prospective or retrospective, clinical or experimental, blind or double-blind. But could you also indicate the inclusion and exclusion criteria for the animals?
Response:
Lines 100-107. I understand that these paragraphs describe the measurement of blood pressure. However, this description raises many questions. In particular, could you indicate what equipment you used to measure blood pressure by oscillometry? Which limb did you use for blood pressure measurement? How many times did you take blood pressure measurements? Although they measured the circumference to determine the type of cuff, could you clarify what size cuff they used? Likewise, could you indicate whether they corrected for mean blood pressure?  
Response:

Lines 108-114. This section is very confusing because it describes the use of invasive methodology in anesthetized animals, when it mentions that the comparison was made in conscious animals. Please clarify the anesthetic management of the animals, the catheterization method, the artery used for catheterization, and whether the catheter used was previously heparinized. 
Response:
Lines 115–120. This section raises the most questions because it mentions that only a Pearson correlation was performed, but it creates a lot of confusion because the title suggests a comparison between animals with and without systemic hypertension. Therefore, I would like to ask how animals with and without hypertension were compared. I suggest that you restructure your statistical analysis to clarify how you made these comparisons. Furthermore, considering that your interest is in validating this method, this analysis is too weak to achieve validation if you did not make a comparison or at least determine the level of sensitivity and specificity, but above all, the main question is: What is the parameter that indicates that a bear has hypertension?
Response:

Line 135. Please only indicate the value of p as “p=0.0001”; it is unclear whether this value is significant or not.
Response:
Line 139. Same comment as above.
Response:
Lines 168-174. I do not understand the purpose of this figure, as it is not cited in the text and has no direct relation to your results. I therefore recommend that it be removed. I invite the authors to only cite figures that are related to their results, as this may confuse the reader.
Response:
Line 192. This idea is correct, but the authors could go into more depth, i.e., why does systolic blood pressure have greater clinical value? Please explain what this means biologically.
Response: 

Lines 195-197. I agree with your idea that this was not the objective of your study, but I invite the authors to be bolder in proposing a cutoff point that indicates that the animal has hypertension. For example, in the case of dogs, it is suggested that a systolic pressure value above 120 mmHg is indicative of systemic hypertension. Based on your results, what parameter do you suggest? This could be a precedent for a study perspective.
Response:
Lines 202–218. This paragraph is isolated, and I do not understand its purpose, as it presents a leap in ideas. I suggest that it be deleted. If the authors allow me, I recommend that they reformulate the idea and deepen the discussion on the biological explanation of the mechanisms that can cause systemic hypertension in these animals, as well as the limitations they found in their study and possible perspectives for further study.
Response:

Reviewer 3 Report

Comments and Suggestions for Authors

It should be noted that the authors have undertaken quite interesting research that may have both practical and applied significance, especially since it concerns a very important aspect: the protection of species or populations threatened with extinction. After reviewing the manuscript, I had some observations. Since the authors state that the manuscript was developed based on medical records, this should be clearly articulated in the Materials and Methods section. It is important to know how many different people performed these measurements and whether they were conducted using the same methodology. Are the results from individual measurements comparable? This is crucial for drawing conclusions and, therefore, for the entire study.

In my opinion, the Discussion section should be significantly revised, as it is not written in accordance with the requirements for this type of chapter in scientific papers. In fact, the entire chapter contains only one reference to the bibliography, the rest being results and conclusions.

Publications no. 9, 10, and 11 from the reference list were not used.

It would be worthwhile to relate our results to other studies in this area, if any, in the discussion. In this case, the conclusions and, if necessary, the summary should be revised.

Conclusion: I suggest clarifying certain issues and making minor corrections to the manuscript (discussion and conclusions) and publishing it, as in my opinion, the results obtained may have practical implications for conservation efforts for this species.

Round 2

Reviewer 1 Report

Comments and Suggestions for Authors

Thank you for incorporating the suggestions. The quality of the manuscript was enhanced.

Author Response

Thank you for spending the time to give our manuscript a second review.

Reviewer 2 Report

Comments and Suggestions for Authors

General comments

I appreciate the authors for considering my comments on their manuscript proposal, which I believe addresses a significant gap in the research area. However, there are still weaknesses that I believe should be addressed beforehand. 

Response:

Particular comments

Line 1. I suggest that you review the types of articles accepted in this journal. In my experience, only “review, original article, or short communication” is accepted. If the authors allow me, this proposal is more similar to an “original article.”
Response:
Lines 9–24. Please consider adjusting the font size and type according to the author guidelines.
Response:

Lines 27- 43. I understand your perspective and concern about not properly justifying your study at this level. However, if the authors allow me, this section should mainly be used as a summary of what you did in your experimental work and, above all, the results of this work. Therefore, I insist and invite the authors to consider my previous comments for this section: These introductory sentences for your abstract are excessive. If the authors allow me, I suggest that they limit themselves to describing the importance of recognizing systemic hypertension in this species as a consequence of its zootechnical purpose.

Response:

Line 34. In response to your question about clarifying my question, I suggest that the authors clarify the objective of their study in this section, as it is not clear what the objective of their study is. Although the objective is described in the introduction, there should be consistency across these sections. In addition, I suggest that they briefly clarify the type of measurement that was performed in their study and, above all, on how many animals. 
Response:

Line 38. I insist, If the authors allow me, they could indicate the demographic characteristics of the animals used. That is, the average age, sex, and average weight of the animals included in their study.

Response:

Line 79. I appreciate your comment and apologize for not being clear with my question. If the authors allow me, I find your observation very interesting, but I suggest adding a few lines about the current lack of information in veterinary medicine. For example, I believe that the main gap is the lack of non-invasive methods that allow for the early detection of systemic hypertension in bears, which could aid in their early treatment.
Response:
Line 86. I appreciate again that you considered the comments on your methodology, but, if the authors allow me, I suggest that in order for the reader to understand this section, you could divide it into animals, blood pressure measurement, experimental design, and statistical analysis. I suggest this recommendation to give greater order to the methodology.

Response:

Figure 5. I insist and urge the authors to be very cautious with the information shown in their results. This figure may be very confusing for the reader, as it is not related to the results of their study. Although I share their view on their example, this can be used in a review article, not an experimental one. I suggest that they only show the results of their study.

Response:

Reviewer 3 Report

Comments and Suggestions for Authors

Thank you for the clarifications and corrections. I think the current version deserves publication.

Author Response

(The authors gave the same response as above.)
